# Delayed Skin Reactions to COVID-19 mRNA-1273 Vaccine: Case Report and Literature Review

**DOI:** 10.3390/vaccines10091412

**Published:** 2022-08-28

**Authors:** Ruei-Lin Wang, Wen-Fang Chiang, Chih-Chiun Chiu, Kuo-An Wu, Chia-Yi Lin, Yung-Hsi Kao, Chih-Pin Chuu, Jenq-Shyong Chan, Po-Jen Hsiao

**Affiliations:** 1Department of Internal Medicine, Taoyuan Armed Forces General Hospital, Taoyuan 325, Taiwan; 2Division of Nephrology, Department of Internal Medicine, Tri-Service General Hospital, National Defense Medical Center, Taipei 114, Taiwan; 3Division of Nephrology, Department of Internal Medicine, Taoyuan Armed Forces General Hospital, Taoyuan 325, Taiwan; 4Division of Infectious Disease, Department of Internal Medicine, Taoyuan Armed Forces General Hospital, Taoyuan 325, Taiwan; 5Division of Infectious Disease and Tropical Medicine, Department of Internal Medicine, Tri-Service General Hospital, National Defense Medical Center, Taipei 114, Taiwan; 6Division of Pulmonary & Critical Care Medicine, Department of Internal Medicine, Taoyuan Armed Forces General Hospital, Taoyuan 325, Taiwan; 7Department of Life Sciences, National Central University, Taoyuan 325, Taiwan; 8Institute of Cellular and System Medicine, National Health Research Institutes, Miaoli County 350, Taiwan; 9Graduate Program for Aging, China Medical University, Taichung 404, Taiwan; 10School of Medicine, Fu-Jen Catholic University, Fu-Jen Catholic University Hospital, New Taipei City 242, Taiwan

**Keywords:** COVID-19, SARS-CoV-2, vaccine, adverse effect, delayed skin reactions, delayed injection site reactions

## Abstract

Background: The COVID-19 mRNA vaccine was granted emergency use authorization (EUA) on December 18, 2020. Some patients experienced a transient, pruritic rash at the injection site, which was referred to as “COVID arm”. It is considered a delayed-type hypersensitivity reaction and occurs mostly in individuals after vaccination with the Moderna vaccine but rarely with other mRNA vaccines. Case Summary: A healthy 33-year-old woman with no history of disease or long-term medication presented with fever and rash on the left upper arm three days after her first vaccination with the mRNA-1273 vaccine (Moderna). Results: After treatment with antihistamines, all lesions gradually resolved over the following 4 to 5 days. Conclusion: We report a case of “COVID arm”: a localized erythematous rash surrounding the injection site that arose three days after the first dose of the Moderna COVID-19 vaccine. Delayed injection site reactions occurred in approximately 0.8% of vaccinated people after the first dose and in approximately 0.2% after the second dose. The lesions persisted for several days and then resolved without treatment. Health care providers were not prepared to address these delayed local reactions to the mRNA-1273 vaccine. Given the scale-up of mass vaccination campaigns worldwide, these skin reactions may likely generate concerns among patients and requests for evaluation. Although these skin reactions have not been consistently recognized, guidance regarding the second dose of the vaccine has varied, and many patients have unnecessarily received antibiotic agents.

## 1. Introduction

A novel coronavirus was identified at the end of 2019 as the cause of a cluster of pneumonia cases in Wuhan, China. It spread rapidly throughout China and the world in a few months. Coronavirus is a member of the family Coronaviridae. “Severe acute respiratory syndrome coronavirus 2 (SARS-CoV-2)” as the name for the new virus was announced by International Committee on Taxonomy of Viruses (ICTV), in order to differentiate from the coronavirus responsible for the SARS outbreak of 2003 [1]. Currently, several coronavirus disease 2019 (COVID-19) vaccines are available, including mRNA vaccines, protein subunit vaccines, vector vaccines, etc. Temporary side effects are common, such as pain at the injection site, fever, headache or feeling tired [2]. Other serious but rare side effects include severe allergic reactions, blood clots, myocarditis, and Guillain–Barre syndrome. mRNA SARS-CoV-2 vaccines have been administered largely in the United States (US) under US Food and Drug Administration emergency use authorization [2]. The benefits of getting vaccinated are much greater than the risks of side effects caused by the vaccine for most people. Regarding clinical manifestations of allergic reactions, the most common symptoms and signs are flushing, itching, urticaria, nasal discharge, nasal congestion, faintness, and syncope [3,4,5,6]. Immediate-type hypersensitivity reactions are caused by degranulation of mast cells. Most reactions are immunoglobulin E (IgE)-mediated. However, other reported allergic reactions to mRNA COVID-19 vaccines include the delayed type [3].

The majority of reported allergic reactions to mRNA COVID-19 vaccines occurred within 30 min of vaccination (mostly within 15 min) [2]. “COVID arm” is an uncommon adverse effect that can present as a transient, localized erythematous rash several days after the first dose of the mRNA-1273 COVID-19 vaccine [4,5,6]. Baden et al. [6] reported that delayed injection site reactions occurred in 244 of the 30,420 participants (0.8%) after the first dose and in 68 participants (0.2%) after the second dose of vaccine administration. In an observational cohort study [4], delayed skin reactions (onset > two days after vaccination) were reported by 1.1% of all females who received the mRNA-1273 vaccine and 2% of females aged 31 to 45 years when stratified by age. In addition, there was no report of delayed skin reactions among the males who received the mRNA-1273 vaccine. This clinical manifestation demonstrated that women were at greater risk of adverse drug reactions than men. The difference between men and women has been proposed including decreased body mass, sex-based differences in health information seeking, sex-based differences in pharmacokinetics and pharmacodynamics. The mechanism of these delayed injection site reactions is not well known, either. The clinical suspicion of delayed-type or T-cell–mediated hypersensitivity is supported by skin biopsy specimens. Previous skin biopsy reports have shown superficial perivascular and perifollicular lymphocytic infiltrates with scattered sporadic mast cells and eosinophils [4,5,6,7]. We presented a case of “COVID arm in a woman”: a localized skin erythematous rash surrounding the injection site that arose three days after the first dose of the Moderna COVID-19 vaccine. This report also made a brief review focused on the incidence, clinical characteristics, and management of delayed skin reactions to COVID-19 mRNA vaccines.

## 2. Case Presentation

The patient was a healthy 33-year-old woman with no history of systemic disease, allergies, or long-term medication. Her family history was unremarkable. She was admitted to the emergency department and presented a pruritic rash on the left upper arm (Figure 1), and fever lasting for one day, on the third day after her first vaccination with the mRNA-1273 vaccine (Moderna) in October 2021. The results of blood laboratory tests, including complete blood cell counts, blood sugar levels, kidney function tests, electrolytes levels, liver function tests, and urinalysis, were all in normal ranges (Table 1). She was treated with oral antihistamines and local corticosteroids. No antibiotics were administered during this period. The prognosis was well, all the lesions gradually resolved over the following 4 to 5 days without other intervention.

## 3. Discussion

Delayed injection site reactions are currently considered delayed-type (Type IV) or T-cell-mediated hypersensitivity, supported by a skin biopsy of one of the late local reactions [7], although it is unclear why the reactions tend not to recur if this is the mechanism. The mRNA vaccines contain polyethylene glycol (PEG), which may cause more inflammation or be more immunogenic. More studies are needed to determine what vaccine component drives these reactions and the immunologic mechanism [8,9,10]. In addition, the delayed-type skin or cutaneous reactions have been reported to be associated with other non-COVID-19 vaccinations [10].

The onset of delayed skin reactions usually begins approximately 3 days after vaccination, while one observational cohort study reported the occurrence 3–9 days (median 7) after the first dose of mRNA-1273 [4]. Jacobson, M.A. et al. reported that 13 women who received the mRNA-1273 vaccine during the first 3 weeks of SARS-CoV-2 vaccine availability presented with a pruritic rash at the injection site appearing 3–9 days after receipt of their initial dose in one hospital in the US [4]. Among these 13 females, 5 women had milder or similar reactions with earlier onset after the second dose. Only one additional woman presented with this delayed reaction after the second dose. No people reported serious adverse events or had severe symptoms that warranted further medical attention. These results suggest that delayed onset pruritic skin rashes at the injection site after mRNA-1273 SARS-CoV-2 vaccine administration lasting from days to weeks occur predominantly in women, usually do not cause serious complications or any sequelae, and should not deter receipt of the second vaccination. Therefore, neither local injection site reactions nor delayed-type hypersensitivity reactions are contraindications to subsequent vaccination. The median onset of cutaneous symptoms after the second dose was earlier than that after the first dose. Patients who have had such reactions should receive their second doses on time, without any special precautions [3]. A trend towards a lower risk of recurrent reactions was observed when the second dose was given on the opposite arm from the first dose [4]. It would be reasonable to suggest that individuals with this reaction after the first dose choose the opposite arm for their second dose. Identified sex differences in reported adverse drug reactions often referred to females. Females generally have a lower lean body mass, reduced hepatic clearance, pharmacodynamic differences in the activity of cytochrome P450 (CYP) enzymes, and metabolize drugs at different rates compared with males. Other important factors include absorption, conjugation, protein binding and renal elimination, which may all have some sex-based differences. However, how these differences lead to an increased risk of delayed skin reactions caused by the COVID-19 vaccine is not very clear [11,12,13,14]. de Vries, S. T et al. also proposed that the differences in adverse drug reactions between females and males may have various origins, including pharmacological and behavioral causes, which need to be further assessed. These differences may eventually result in sex-specific prescribing and monitoring recommendations [15]. However, the genetic effects associated with or enhance the adverse reactions after the vaccinations need to be further assessed [14,15]. We also reviewed the literature and compared clinical features, treatment, and clinical outcomes in patients with COVID-19 mRNA-1273 vaccine-induced delayed skin reactions (Table 2).

In summary, current clinical data suggest that potential recipients of the mRNA-1273 vaccine can be reassured that delayed injection site skin reactions are benign, self-limited adverse events that resolve gradually and do not lead to serious clinical problems. It is important for public health agencies to educate the public about the possibility of these delayed injection site reactions to the mRNA-1273 vaccine and reassure them. Nevertheless, more evidence is needed to characterize the epidemiology, mechanism, prevention, treatment, and why delayed injection site reactions seems to occur more frequently in females than males, even among different races.

## 4. Conclusions

We reported a case of a delayed skin injection site reaction to the mRNA-1273 vaccine. The comprehensive mechanism is not well known, and the clinical characteristics are consistent with delayed-type hypersensitivity.

## Figures and Tables

**Figure 1 vaccines-10-01412-f001:**
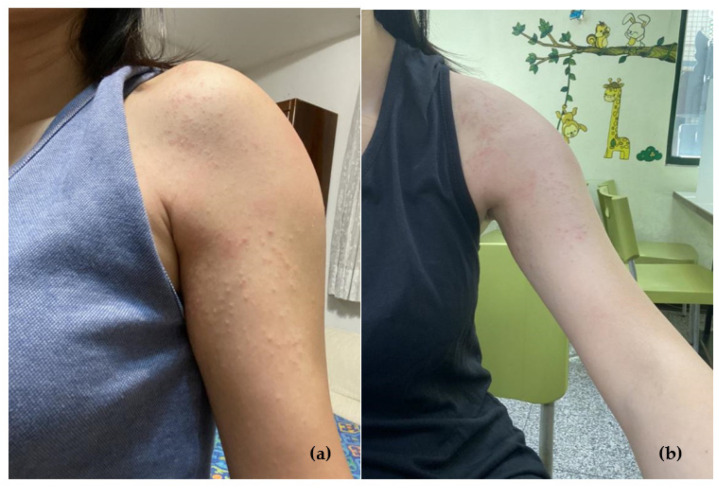
(**a**) Erythematous rash over the left upper arm two days after receiving the mRNA-1273 vaccine. (**b**) The arm four days later after antihistamine treatment.

**Table 1 vaccines-10-01412-t001:** Results of blood biochemistry tests and complete blood cell counts.

Parameter	Result	Unit	Normal Range
BUN	18.2	mg/dL	7–25
Creatinine	0.60	mg/dL	F: 0.44–1.03; M: 0.64–1.27
eGFR	82.5	ml/min/1.732 m^2^	
Sodium	138.2	mmol/L	136–146
Potassium	4.3	mmol/L	3.5–5.1
Calcium	8.8	mg/dL	8.6–10.3
Chloride	106.8	mmol/L	101–109
GOT	25	mmol/L	Adult: ≤34
GPT	30	mmol/L	Adult: ≤36
Total bilirubin	0.92	mmol/L	0.3–1.2 (5 days–60 y)
Glucose	88.7	mg/dL	AC: 74–100 (≥18 y) PC: <140 (≥18 y)
White blood cell count	9.05	$10^{3}$/µl	M: 3.9–10.6; F: 3.5–11
Red blood cell count	4.0	$10^{6}$/µl	M: 4.5–5.9; F: 4.0–5.2
Hemoglobin	13.1	g/dL	M: 13.5–17.5; F: 12–16
Hematocrit	40.2	%	M: 41–53; F36–46
MCV	96.8	fl	80–100
MCH	31.7	pg	26–34
MCHC	32.5	g/dL	31–37
Platelet count	203	$10^{3}$/mm	150–400

BUN: blood urea nitrogen; eGFR: estimated glomerular filtration rate; GOT: glutamyl oxaloacetic transaminase; GPT: glutamyl pyruvate transaminase; MCV: mean corpuscular volume; MCH: mean corpuscular hemoglobin; and MCHC: mean corpuscular hemoglobin concentration.

**Table 2 vaccines-10-01412-t002:** Literature reviews of COVID-19 mRNA-1273 vaccine-induced delayed skin reactions.

Ref.	Clinical Feature	Case Number/ Duration after Vaccination	Relevant Data or Examination	Treatment	Resolving Date after Onset
[3]	Plaques, fatigue, headache.	12. 4–11 days (median on day 8).	Skin biopsy showed superficial perivascular and perifollicular lymphocytic infiltrates.	Steroids, antibiotic treatment.	2–11 days (median on day 6).
[4]	Erythema, pruritus, tenderness.	13. 3–9 days.	Skin biopsy demonstrated a largely perivascular lymphocytic infiltrate with little epidermal change.	Steroids, antihistamine.	Within a week.
[5]	Pruritic erythematous rash, maculopapular eruption.	4. 7–10 days.	Not mentioned.	Steroids, antihistamine.	Not mentioned.
[6]	Erythema, induration, and tenderness.	244. 8 days or more.	Not mentioned.	Not mentioned.	4–5 days.
[7]	Pain, with or without associated redness and swelling.	12. One week.	Not mentioned.	Steroids, antihistamine, ice.	3–8 days.

## Data Availability

The data underlying this article will be shared upon reasonable request to the corresponding author.

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
