# Peer review of "Delayed Skin Reactions to COVID-19 mRNA-1273 Vaccine: Case Report and Literature Review"

_vaccines, 2022, doi:10.3390/vaccines10091412_

Round 1
Reviewer 1 Report
This communication review provides interesting information on the rare adverse effects after COVID-19 vaccination. Here a case of “COVID arm” is reported: a localized erythematous rash surrounding the injection site that appeared three days after the first dose of the Moderna COVID-19 vaccine.
Comment:
1. Delayed injection site reactions occurred in approximately 0.8% of vaccinated people after the first dose and in approximately 0.2% after the second dose. All clinical practice guidelines admit that neither local injection site reactions nor delayed-type hypersensitivity reactions are contraindications to subsequent vaccination. Does this new case that is now reported have some special characteristics?. Would it be useful to highlight what knowledge the description of this case provides?2. A fairly brief review of published cases is provided. It could be interesting to expand the information and include a summary table with the main descriptions of the cases described.
Author Response
Response to Reviewer 1
[General Comment]
This communication review provides interesting information on the rare adverse effects after COVID-19 vaccination. Here a case of “COVID arm” is reported: a localized erythematous rash surrounding the injection site that appeared three days after the first dose of the Moderna COVID-19 vaccine.
Author Reply: We sincerely appreciate your time and effort spent reviewing this manuscript. We have revised the manuscript thoroughly according to all reviewer’s suggestions. The responses to your comments are found below. All changes have been highlighted with yellowish colors.
Comments:
- Delayed injection site reactions occurred in approximately 0.8% of vaccinated people after the first dose and in approximately 0.2% after the second dose. All clinical practice guidelines admit that neither local injection site reactions nor delayed-type hypersensitivity reactions are contraindications to subsequent vaccination. Does this new case that is now reported have some special characteristics? Would it be useful to highlight what knowledge the description of this case provides?
Author reply: We sincerely appreciate your time and effort spent reviewing this manuscript. We have revised the manuscript thoroughly according to your suggestions. Owing to this case, we identified the possible skin adverse side effects after the COVID-19 vaccinations in the clinical scenario. Furthermore, we learned more about clinical presentation and appropriate management. Clinicians could be more aware of delayed skin reactions to the COVID-19 vaccines or even other vaccines, and give proper treatment to the patients who encounter the same clinical situations. Please see the section of Discussion.
- A fairly brief review of published cases is provided. It could be interesting to expand the information and include a summary table with the main descriptions of the cases described.
Author reply: We sincerely appreciate your valuable comment. We have revised the manuscript thoroughly according to your suggestions. We have added the literature review (Table 3) in the revised manuscript.
Table 3. Literature review of COVID-19 mRNA-1273 vaccine-induced skin reactions.
|
Ref. |
Clinical feature |
Case number/ Duration after vaccination |
Relevant data or examination |
Treatment |
Resolving date after onset |
|
[1] |
Plaques, fatigue, headache. |
12. 4-11 days (median on day 8). |
Skin biopsy showed superficial perivascular and perifollicular lymphocytic infiltrates. |
Steroids, antibiotic treatment. |
2-11 days (median on day 6). |
|
[4] |
Erythema, pruritus, tenderness. |
13. 3-9 days. |
Skin biopsy demonstrated a largely perivascular lymphocytic infiltrate with little epidermal change. |
Steroids, antihistamine. |
Withing a week. |
|
[5] |
Pruritic erythematous rash, maculopapular eruption. |
4. 7-10 days. |
Not mentioned. |
Steroids, antihistamine. |
Not mentioned. |
|
[6] |
Erythema, induration, and tenderness. |
244. 8 days or more. |
Not mentioned. |
Not mentioned. |
4-5 days. |
|
[7] |
Pain, with or without associated redness and swelling. |
12. One week. |
Not mentioned. |
Steroids, antihistamine, ice. |
3-8 days. |
Last, we are deeply honored by the time and effort you spent reviewing this manuscript. In reviewing and revising our manuscript, we are motivated to read more and thus learn more from your criticisms.

Reviewer 2 Report
The article entitled “ Incidence, Clinical Characteristics, and Management of Delayed Skin Reactions to mRNA COVID-19 Vaccines: A Case Report and Literature Review”, presented data on allergic reaction after administration of mRNA vaccine to a healthy woman and short review on literature on closely related allergic reactions to these vaccine.
In general: type of article is “case report”. So, it should be presented as a case report. Usually the section of “introduction” shows a short “review”, or, it means, introduction: what it is the subject. In this case: 1. Vaccines ( different types, what is specific in mRNA vaccines, risks(?)), 2 . allergic reactions. The “ body of the article”: clinical signs, treatment, prognoses(?) of so. Discussion: what is new comparing to already known. I suggest to change the title of the article.
Lines 35-37. “ Delayed injection site reactions occurred in approximately 0.8% of vaccinated people after the first dose and in approximately 0.2% after the second dose.” I suggest adding the source of information.
Introduction.
In general. A common information on the coronavirus (nomenclature) should be added.
Line 58 “risks for most people” . I suggest to add “side effects caused by vaccine”
Lines 62-64. “However, other reported reactions to mRNA COVID-19 vaccines include delayed types. We present a case of a delayed injection site reaction to the mRNA-1273 vaccine against SARS-CoV-2.” “Allergic reaction” is lacking in these two sentences and have to be added.
Lines 67-68. “The second vaccination is suggested to be administered on time, even though patients may have recurrent reactions similar to those after the first vaccination”. Do authors mean a local reaction? If so, it should be proved accordingly.
Line 72. How long lasted fever?
Lines 81-83. Reference has to be added.
Line 87. “after the second dose.” “of vaccine administration” have to be added.
Line 90 “age,.” Punctuation
Lines 87-93. The order of sentences should be changed for consequent and clear story.
Line 93-95. Hypothesis is written not clear.
Lines 81-95. In my view, most data from this paragraph should be moved to the introduction section. Discussion section should include discussion (comparison, summery, conclusions) and minimum presented data.
Lines 96-105. The same.
Table 1. Whether is the table showing data related to allergic reaction caused by vaccine or not? If not, I don’t believe that it isn’t necessary for this article and has to be deleted.
Conclusion.
This section should not include recommendation. It is only a case report, based on a single case. Treatment of the patient should me moved to “ Case presentation “ section.
Lines 155-158. Useless information. To delete.

Author Response
Response to Reviewer 2
[General Comment]
The article entitled “Incidence, Clinical Characteristics, and Management of Delayed Skin Reactions to mRNA COVID-19 Vaccines: A Case Report and Literature Review”, presented data on allergic reaction after administration of mRNA vaccine to a healthy woman and short review on literature on closely related allergic reactions to these vaccines.
Author Reply: We sincerely appreciate your time and effort spent reviewing this manuscript. We have revised the manuscript thoroughly according to all reviewer’s suggestions. The responses to your comments are found below. All changes have been highlighted with yellowish colors.
Comments:
In general: type of article is “case report”. So, it should be presented as a case report. Usually the section of “introduction” shows a short “review”, or, it means, introduction: what it is the subject. In this case: 1. Vaccines (different types, what is specific in mRNA vaccines, risks(?)), 2. allergic reactions. The “body of the article”: clinical signs, treatment, prognoses(?) of so. Discussion: what is new comparing to already known. I suggest to change the title of the article.
Author reply: We sincerely appreciate your time and effort spent reviewing this manuscript. We have revised the manuscript thoroughly according to your suggestions. The responses to your comments are found below. The subject was that we presented a case of a delayed localized skin reaction to the mRNA-1273 vaccine against SARS-CoV-2. The cutaneous reactions include rash, erythema, induration, pruritus, and tenderness in this patient. The patient was a healthy 33-year-old woman with no history of systemic disease, allergies, or long-term medication. Her family history was unremarkable. She presented a pruritic rash on the left upper arm (Figure 1), and fever lasting for one day, on the third day after her first vaccination with the mRNA-1273 vaccine (Moderna). The results of blood laboratory tests, including complete blood cell counts, blood sugar levels, kidney function tests, electrolytes levels, liver function tests, and urinalysis, were all in normal ranges (Table 1). She was treated with oral antihistamines and local corticosteroids. No antibiotics were administered during this period. The prognosis was well, all the lesions gradually resolved over the following 4 to 5 days without other intervention.
Owing to this case, we identified the possible skin adverse side effects after the COVID-19 vaccinations in the clinical scenario. Furthermore, we learned more about clinical presentation and appropriate management. Clinicians could be more aware of delayed skin reactions to the COVID-19 vaccines or even other vaccines, and give proper treatment to the patients who encounter the same clinical situations. Please see the section of Discussion.
Finally, we also revised the title of our manuscript to “Delayed Skin Reactions to COVID-19 mRNA-1273 Vaccine: Case Report and Literature Review”
Lines 35-37. “Delayed injection site reactions occurred in approximately 0.8% of vaccinated people after the first dose and in approximately 0.2% after the second dose.” I suggest adding the source of information.
Author reply: Thanks for your comment. We have added the reference [1].
Reference
- Blumenthal, K.G.; Freeman, E.E.; Saff, R.R.; Robinson, L.B.; Wolfson, A.R.; Foreman, R,K. Hashimoto, D.; Banerji, A.; Li, L.; Anvari, S.; Shenoy, E.S. Delayed Large Local Reactions to mRNA-1273 Vaccine against SARS-CoV-2. N Engl J Med 2021, 384, 1273-1277.
Introduction.
In general. A common information on the coronavirus (nomenclature) should be added.
Author reply: We sincerely appreciate your valuable comment. We have added the nomenclature and reference 2 in the manuscript. Coronavirus is a member of the family Coronaviridae. “Severe acute respiratory syndrome coronavirus 2 (SARS-CoV-2)” as the name for the new virus was announced by International Committee on Taxonomy of Viruses (ICTV), in order to differentiate from the coronavirus responsible for the SARS outbreak of 2003 [2]. Please see the section of the Introduction.
Reference
- Khan, T., Jamal, S.M. SARS-CoV-2 nomenclature: viruses, variants and vaccines need a standardized naming system. Future Virol 2021 Oct, 10.2217/fvl-2021-0198. doi: 10.2217/fvl-2021-0198.
Line 58 “risks for most people”. I suggest to add “side effects caused by vaccine”
Author reply: Thanks for your valuable comment. We have revised according to your suggestion. We have revised this sentence to “The benefits of getting vaccinated are much greater than the risks of side effects caused by vaccine for most people.” Please see the section of Introduction.
Lines 62-64. “However, other reported reactions to mRNA COVID-19 vaccines include delayed types. We present a case of a delayed injection site reaction to the mRNA-1273 vaccine against SARS-CoV-2.” “Allergic reaction” is lacking in these two sentences and have to be added.
Author reply: Thanks for your valuable comment. We have revised this sentence to “However, other reported allergic reactions to mRNA COVID-19 vaccines include the delayed type [1].” Please see the section of Introduction.
Lines 67-68. “The second vaccination is suggested to be administered on time, even though patients may have recurrent reactions similar to those after the first vaccination”. Do authors mean a local reaction? If so, it should be proved accordingly.
Author reply: Thanks for your valuable comment. We have revised this sentence to “The second vaccination could be suggested to be administered on time, even though patients may have recurrent delayed localized skin reactions similar to those after the first vaccination.” Please see the section of Introduction.
Line 72. How long lasted fever?
Author reply: Thanks for your valuable comment. Her family history was unremarkable. She presented a pruritic rash on the left upper arm (Figure 1), and fever lasting for one day, on the third day after her first vaccination with the mRNA-1273 vaccine (Moderna).
Lines 81-83. Reference has to be added.
Author reply: Thanks for your valuable comment. The cause of allergic or anaphylactic reactions to COVID-19 vaccines has yet to be determined. The majority of reported allergic reactions to mRNA COVID-19 vaccines occurred within 30 minutes of vaccination (mostly within 15 minutes) [3]
Reference
- CDC COVID-19 Response Team; Food and Drug Administration. Allergic Reactions Including Anaphylaxis After Receipt of the First Dose of Moderna COVID-19 Vaccine — United States, December 21, 2020–January 10, 2021. MMWR Morb Mortal Wkly Rep 2021;70:125–129.
Line 87. “after the second dose.” “of vaccine administration” have to be added.
Author reply: Thanks for your comment. We have revised as suggestion. Please see the section of Discussion.
Line 90 “age,.” Punctuation
Author reply: Thanks for your comment. We have made the correction of typographical error.
Lines 87-93. The order of sentences should be changed for consequent and clear story.
Author reply: Thanks for your comment. We have revised this sentence to “In an observational cohort study [2], delayed skin reactions (onset > two days after vaccination) were reported by 1.1% of all female who received the mRNA-1273 vaccine and 2% of females aged 31 to 45 years when stratified by age. In addition, there were no report of delayed skin reactions among the male who received the mRNA-1273 vaccine. This study results demonstrated that women were at greater risk of adverse drug reactions than men.” Please see the section of Discussion.
Line 93-95. Hypothesis is written not clear.
Author reply: Thanks for your valuable comment. We have revised this sentence to “The difference between men and women has been proposed including decreased body mass, sex-based differences in health information seeking, sex-based differences in pharmacokinetics and pharmacodynamics.” Please see the section of Discussion.
Lines 81-95. In my view, most data from this paragraph should be moved to the introduction section. Discussion section should include discussion (comparison, summery, conclusions) and minimum presented data.
Author reply: We sincerely appreciate your valuable comment. We have revised according to your suggestions.
- Introduction
A novel coronavirus was identified at the end of 2019 as the cause of a cluster of pneumonia cases in Wuhan, China. It spread rapidly throughout China and the world in a few months. Coronavirus is a member of the family Coronaviridae. "Severe acute respiratory syndrome coronavirus 2 (SARS-CoV-2)" as the name for the new virus was announced by International Committee on Taxonomy of Viruses (ICTV), in order to differentiate from the coronavirus responsible for the SARS outbreak of 2003 [2]. Currently, several coronavirus disease 2019 (COVID-19) vaccines are available, including mRNA vaccines, protein subunit vaccines, vector vaccines, etc. Temporary side effects are common, such as pain at the injection site, fever, headache or feeling tired. Other serious but rare side effects include severe allergic reactions, blood clots, myocarditis, and Guillain–Barre syndrome. mRNA SARS-CoV-2 vaccines have been administered largely in the United States (US) under US Food and Drug Administration emergency use authorization. The benefits of getting vaccinated are much greater than the risks of side effects caused by vaccine for most people. Regarding clinical manifestations of allergic reactions, the most common symptoms and signs are flushing, itching, urticaria, nasal discharge, nasal congestion, faintness, and syncope. Immediate-type hypersensitivity reactions are caused by degranulation of mast cells. Most reactions are immunoglobulin E (IgE)-mediated. However, other reported allergic reactions to mRNA COVID-19 vaccines include the delayed type [1]. The subject was that we presented a case of a delayed localized skin reaction to the mRNA-1273 vaccine against SARS-CoV-2. The cutaneous reactions include rash, erythema, induration, pruritus, and tenderness in this patient. Although the skin reactions may be of concern to this patient and the clinicians, these adverse side effects are self-limited and typically resolve over the following 5 days. After the 3-month later, she received the 2nd dose of COVID-19 mRNA-1273 vaccination, no obvious skin delayed reactions were noted again. The second vaccination could be suggested to be administered on time, even though patients may have recurrent delayed localized skin reactions similar to those after the first vaccination. This report also made a brief review focused on the incidence, clinical characteristics, and management of delayed skin reactions to mRNA COVID-19 vaccines.
Lines 96-105. The same.
Author reply: We sincerely appreciate your valuable comment. We have revised the content in the section of Introduction.
- Introduction
A novel coronavirus was identified at the end of 2019 as the cause of a cluster of pneumonia cases in Wuhan, China. It spread rapidly throughout China and the world in a few months. Coronavirus is a member of the family Coronaviridae. "Severe acute respiratory syndrome coronavirus 2 (SARS-CoV-2)" as the name for the new virus was announced by International Committee on Taxonomy of Viruses (ICTV), in order to differentiate from the coronavirus responsible for the SARS outbreak of 2003 [2]. Currently, several coronavirus disease 2019 (COVID-19) vaccines are available, including mRNA vaccines, protein subunit vaccines, vector vaccines, etc. Temporary side effects are common, such as pain at the injection site, fever, headache or feeling tired. Other serious but rare side effects include severe allergic reactions, blood clots, myocarditis, and Guillain–Barre syndrome. mRNA SARS-CoV-2 vaccines have been administered largely in the United States (US) under US Food and Drug Administration emergency use authorization. The benefits of getting vaccinated are much greater than the risks of side effects caused by vaccine for most people. Regarding clinical manifestations of allergic reactions, the most common symptoms and signs are flushing, itching, urticaria, nasal discharge, nasal congestion, faintness, and syncope. Immediate-type hypersensitivity reactions are caused by degranulation of mast cells. Most reactions are immunoglobulin E (IgE)-mediated. However, other reported allergic reactions to mRNA COVID-19 vaccines include the delayed type [1]. The subject was that we presented a case of a delayed localized skin reaction to the mRNA-1273 vaccine against SARS-CoV-2. The cutaneous reactions include rash, erythema, induration, pruritus, and tenderness in this patient. Although the skin reactions may be of concern to this patient and the clinicians, these adverse side effects are self-limited and typically resolve over the following 5 days. After the 3-month later, she received the 2nd dose of COVID-19 mRNA-1273 vaccination, no obvious skin delayed reactions were noted again. The second vaccination could be suggested to be administered on time, even though patients may have recurrent delayed localized skin reactions similar to those after the first vaccination. This report also made a brief review focused on the incidence, clinical characteristics, and management of delayed skin reactions to mRNA COVID-19 vaccines.
Table 1. Whether is the table showing data related to allergic reaction caused by vaccine or not? If not, I don’t believe that it isn’t necessary for this article and has to be deleted.
Author reply: We sincerely appreciate your valuable comment. Depending on available references with skin biopsy, delayed skin reactions to COVID-19 mRNA-1273 vaccine was considered as delayed-type (Type IV) or T-cell-mediated hypersensitivity. We provided basic data of different types of hypersensitivity (Table 2) for the purpose of making readers clearer about differences between varied clinical presentations.
Conclusion.
This section should not include recommendation. It is only a case report, based on a single case. Treatment of the patient should me moved to “Case presentation “section.
Author reply: We sincerely appreciate your valuable comment. Treatment of the patient has been removed to “Case presentation “section.
Lines 155-158. Useless information. To delete.
Author reply: We sincerely appreciate your valuable comment. We have deleted this immediately. We have revised the content in the section of Conclusions.
Last, we are deeply honored by the time and effort you spent reviewing this manuscript. In reviewing and revising our manuscript, we are motivated to read more and thus learn more from your criticisms.

Reviewer 3 Report
This manuscript by Wang et al., reports a delayed case of pruritic rash in a female patient after receiving the mRNA COVID-19 vaccine (Moderna) and suitable management strategy. It is suitable for publication in this journal after minor revision. I have some minor questions that needs to be addressed first:
1. I would like to ask the authors to add more detailed information regarding the patient’s check-in in the hospital, such as what time (at least which season, etc), the patient’s basic parameters as well as result of blood tests if she was asked to do so.
2. For the delayed pruritic rash, is there any report regrading whether it can be happened in patients with injection of any non-COVID vaccines or other treatment or in healthy people with seasonal weather change? The authors showed that it is mostly happened in patients with Moderna vaccination.
3. The authors discussed about gender and also racial difference in this manuscript. If there is any genetic mutations associated with or enhance it after vacation, please add it in the discussion.
Thanks for the invitation.
Author Response
Response to Reviewer 3
[General Comment]
This manuscript by Wang et al., reports a delayed case of pruritic rash in a female patient after receiving the mRNA COVID-19 vaccine (Moderna) and suitable management strategy. It is suitable for publication in this journal after minor revision. I have some minor questions that needs to be addressed first:
Author Reply: We sincerely appreciate your time and effort spent reviewing this manuscript. We have revised the manuscript thoroughly according to all reviewer’s suggestions. The responses to your comments are found below. All changes have been highlighted with yellowish colors.
Minor Comments:
- I would like to ask the authors to add more detailed information regarding the patient’s check-in in the hospital, such as what time (at least which season, etc), the patient’s basic parameters as well as result of blood tests if she was asked to do so.
Author reply: We sincerely appreciate your valuable comment. She was admitted to the emergency department and presented a pruritic rash on the left upper arm (Figure 1), and fever lasting for one day, on the third day after her first vaccination with the mRNA-1273 vaccine (Moderna) in October, 2021. The results of blood laboratory tests, including complete blood cell counts, blood sugar levels, kidney function tests, electrolytes levels, liver function tests, and urinalysis, were all in normal ranges (Table 1).
Table 1. Results of blood biochemistry tests and complete blood cell counts
|
Parameter |
Result |
Unit |
Normal Range |
|
BUN |
18.2 |
mg/dL |
7–25 |
|
Creatinine |
0.60 |
mg/dL |
F: 0.44–1.03; M: 0.64–1.27 |
|
eGFR |
82.5 |
ml/min/1.732 m2 |
|
|
Sodium |
138.2 |
mmol/L |
136–146 |
|
Potassium |
4.3 |
mmol/L |
3.5–5.1 |
|
Calcium |
8.8 |
mg/dL |
8.6–10.3 |
|
Chloride |
106.8 |
mmol/L |
101–109 |
|
GOT |
25 |
mmol/L |
Adult: ≤34 |
|
GPT |
30 |
mmol/L |
Adult: ≤36 |
|
Total bilirubin |
0.92 |
mmol/L |
0.3–1.2 (5 days-60 y) |
|
Glucose |
88.7 |
mg/dL |
AC: 74–100 (≥18 y) PC: <140 (≥18 y) |
|
White blood cell count |
9.05 |
/µl |
M: 3.9–10.6; F: 3.5–11 |
|
Red blood cell count |
4.0 |
/µl |
M: 4.5–5.9; F: 4.0–5.2 |
|
Hemoglobin |
13.1 |
g/dL |
M: 13.5–17.5; F: 12–16 |
|
Hematocrit |
40.2 |
% |
M: 41–53; F36–46 |
|
MCV |
96.8 |
fl |
80–100 |
|
MCH |
31.7 |
pg |
26–34 |
|
MCHC |
32.5 |
g/dl |
31–37 |
|
Platelet count |
203 |
/mm |
150–400 |
BUN: blood urea nitrogen; eGFR: estimated glomerular filtration rate; GOT: glutamyl oxaloacetic transaminase; GPT: glutamyl pyruvate transaminase; MCV: mean corpuscular volume; MCH: mean corpuscular hemoglobin; and MCHC: mean corpuscular hemoglobin concentration.
- For the delayed pruritic rash, is there any report regrading whether it can be happened in patients with injection of any non-COVID vaccines or other treatment or in healthy people with seasonal weather change? The authors showed that it is mostly happened in patients with Moderna vaccination.
Author reply: We sincerely appreciate your valuable comment. In addition, the delayed-type skin or cutaneous reactions have been reported to be associated with other non-COVID-19 vaccinations [10]. We have added this point in our manuscript. Please see the section of Discussion.
Reference
- McNeil, M.M.; DeStefano, F. Vaccine-associated hypersensitivity. J Allergy Clin Immunol 2018, 141, 463–472. doi: 10.1016/j.jaci.2017.12.971.
- The authors discussed about gender and also racial difference in this manuscript. If there is any genetic mutations associated with or enhance it after vacation, please add it in the discussion.
Author reply: We sincerely appreciate your valuable comment. It would be reasonable to suggest that individuals with this reaction after the first dose choose the opposite arm for their second dose. Identified sex differences in reported adverse drug reactions often referred to females. Females generally have a lower lean body mass, reduced hepatic clearance, pharmacodynamic differences in the activity of cytochrome P450 (CYP) enzymes, and metabolize drugs at different rates compared with males. Other important factors include absorption, conjugation, protein binding and renal elimination, which may all have some sex-based differences. However, how these differences lead to an increased risk of delayed skin reactions caused by the COVID-19 vaccine is not very clear [11-14].
de Vries, S. T et al. also proposed that the differences in adverse drug reactions between females and males may have various origins, including pharmacological and behavioural causes, which need to be further assessed. These differences may eventually result in sex-specific prescribing and monitoring recommendations [13]. However, the genetic effects associated with or enhance the adverse reactions after the vaccinations need to be further assessed [14,15].
Last, we are deeply honored by the time and effort you spent reviewing this manuscript. In reviewing and revising our manuscript, we are motivated to read more and thus learn more fr

Round 2
Reviewer 1 Report
The discussion of the case presented has improved with the changes introduced. Thank you very much
Author Response
Response to Reviewer 1
[General Comment]
The discussion of the case presented has improved with the changes introduced. Thank you very much
Author Reply: We sincerely appreciate your time and effort spent reviewing this manuscript. We have revised the manuscript thoroughly according to all reviewer’s suggestions. All changes have been highlighted with yellowish colors.
Last, we are deeply honored by the time and effort you spent reviewing this manuscript. In reviewing and revising our manuscript, we are motivated to read more and thus learn more from your criticisms.
Reviewer 2 Report
The article entitled” Delayed Skin Reactions to COVID-19 mRNA-1273 Vaccine: Case Report and Literature Review“ presented data on allergic reaction after administration of mRNA vaccine to a healthy woman and short review on literature on closely related allergic reactions to these vaccine.
In general. Authors made changes on all small remarks. Unfortunately, structure of the case report was not complied. A case report should include description of a clinical case, as a main issue. Introduction section is still insufficient, whereas the discussion section provides a lot of abundant data useless for a regular case report.
Line 36. In abstract reference has not being shown.
Author Response
Response to Reviewer 2
[General Comment]
The article entitled” Delayed Skin Reactions to COVID-19 mRNA-1273 Vaccine: Case Report and Literature Review “presented data on allergic reaction after administration of mRNA vaccine to a healthy woman and short review on literature on closely related allergic reactions to these vaccine.
Author Reply: We sincerely appreciate your time and effort spent reviewing this manuscript. We have revised the manuscript thoroughly according to all reviewer’s suggestions. The responses to your comments are found below. All changes have been highlighted with yellowish colors.
Comments:
In general. Authors made changes on all small remarks. Unfortunately, structure of the case report was not complied. A case report should include description of a clinical case, as a main issue. Introduction section is still insufficient, whereas the discussion section provides a lot of abundant data useless for a regular case report.
Author reply: We sincerely appreciate your time and effort spent reviewing this manuscript. We revised the manuscript by adjusting contents of introduction and discussion sections to achieve your suggestions.
- Introduction
A novel coronavirus was identified at the end of 2019 as the cause of a cluster of pneumonia cases in Wuhan, China. It spread rapidly throughout China and the world in a few months. Coronavirus is a member of the family Coronaviridae. "Severe acute respiratory syndrome coronavirus 2 (SARS-CoV-2)" as the name for the new virus was announced by International Committee on Taxonomy of Viruses (ICTV), in order to differentiate from the coronavirus responsible for the SARS outbreak of 2003 [2]. Currently, several coronavirus disease 2019 (COVID-19) vaccines are available, including mRNA vaccines, protein subunit vaccines, vector vaccines, etc. Temporary side effects are common, such as pain at the injection site, fever, headache or feeling tired. Other serious but rare side effects include severe allergic reactions, blood clots, myocarditis, and Guillain–Barre syndrome. mRNA SARS-CoV-2 vaccines have been administered largely in the United States (US) under US Food and Drug Administration emergency use authorization. The benefits of getting vaccinated are much greater than the risks of side effects caused by vaccine for most people. Regarding clinical manifestations of allergic reactions, the most common symptoms and signs are flushing, itching, urticaria, nasal discharge, nasal congestion, faintness, and syncope. Immediate-type hypersensitivity reactions are caused by degranulation of mast cells. Most reactions are immunoglobulin E (IgE)-mediated. However, other reported allergic reactions to mRNA COVID-19 vaccines include the delayed type [1]. The subject was that we presented a case of a delayed localized skin reaction to the mRNA-1273 vaccine against SARS-CoV-2. The cutaneous reactions include rash, erythema, induration, pruritus, and tenderness in this patient. Although the skin reactions may be of concern to this patient and the clinicians, these adverse side effects are self-limited and typically resolve over the following 5 days. After the 3-month later, she received the 2nd dose of COVID-19 mRNA-1273 vaccination, no obvious skin delayed reactions were noted again. The second vaccination could be suggested to be administered on time, even though patients may have recurrent delayed localized skin reactions similar to those after the first COVID-19 mRNA-1273 vaccination.
The cause of allergic or anaphylactic reactions to COVID-19 vaccines has yet to be determined. The majority of reported allergic reactions to mRNA COVID-19 vaccines occurred within 30 minutes of vaccination (mostly within 15 minutes) [3]. "COVID arm" is an uncommon adverse effect that can present as transient, localized erythematous rash several days after the first dose of the mRNA-1273 COVID-19 vaccine [4-6]. Baden et al. [6] reported that delayed injection site reactions occurred in 244 of the 30,420 participants (0.8%) after the first dose and in 68 participants (0.2%) after the second dose of vaccine administration. In an observational cohort study [4], delayed skin reactions (onset > two days after vaccination) were reported by 1.1% of all female who received the mRNA-1273 vaccine and 2% of females aged 31 to 45 years when stratified by age. In addition, there were no report of delayed skin reactions among the male who received the mRNA-1273 vaccine. This clinical manifestation demonstrated that women were at greater risk of adverse drug reactions than men. The difference between men and women has been proposed including decreased body mass, sex-based differences in health information seeking, sex-based differences in pharmacokinetics and pharmacodynamics. The mechanism of these delayed injection site reactions is not well known, either. The clinical suspicion of delayed-type or T-cell–mediated hypersensitivity is supported by skin biopsy specimens. Previous skin biopsy reports have shown superficial perivascular and perifollicular lymphocytic infiltrates with scattered sporadic mast cells and eosinophils [4-7]. We presented a case of “COVID arm in a woman”: a localized skin erythematous rash surrounding the injection site that arose three days after the first dose of the Moderna COVID-19 vaccine. This report also made a brief review focused on the incidence, clinical characteristics, and management of delayed skin reactions to COVID-19 mRNA vaccines.
Line 36. In abstract reference has not being shown.
Author reply: We had added the reference [1]. If we misunderstood your advice, please enlighten us.
Reference
- Blumenthal, K.G.; Freeman, E.E.; Saff, R.R.; Robinson, L.B.; Wolfson, A.R.; Foreman, R,K. Hashimoto, D.; Banerji, A.; Li, L.; Anvari, S.; Shenoy, E.S. Delayed Large Local Reactions to mRNA-1273 Vaccine against SARS-CoV-2. N Engl J Med 2021, 384, 1273-1277.
Last, we are deeply honored by the time and effort you spent reviewing this manuscript. In reviewing and revising our manuscript, we are motivated to read more and thus learn more from your criticisms.
Round 3
Reviewer 2 Report
The article entitled” Delayed Skin Reactions to COVID-19 mRNA-1273 Vaccine: Case Report and Literature Review“ presented data on allergic reaction after administration of mRNA vaccine to a healthy woman and short review on literature on closely related allergic reactions to these vaccine.
Major. The authors describe the case four times: in introduction, body, discussion and conclusion. The data has to be presented at the “body” text only. In all other part of the manuscript have to be significantly shortened.
Lines 63-73. This data should be present in the “body” of the article only. To delete.
Discussion. Too long and boring. Too much abundant data.
Conclusions. In my opinion, basing on a single case the authors cannot made any conclusions. It should be completely deleted.
Minor.
Lines 51-63. Reference to every statement has to be added.
Author Response
Response to Reviewer 2
[General Comment]
The article entitled” Delayed Skin Reactions to COVID-19 mRNA-1273 Vaccine: Case Report and Literature Review“ presented data on allergic reaction after administration of mRNA vaccine to a healthy woman and short review on literature on closely related allergic reactions to these vaccine.
Author Reply: We sincerely appreciate your time and effort spent reviewing this manuscript. We have revised the manuscript thoroughly according to your suggestions. The responses to your comments are found below.
Major. The authors describe the case four times: in introduction, body, discussion and conclusion. The data has to be presented at the “body” text only. In all other part of the manuscript have to be significantly shortened.
Author Reply:
Thank you for your valuable comments. We have made this correction immediately. We made the shortening in Introduction, Discussion and Conclusion. In addition, we deleted the original Table 2 according to your previous suggestion.
- Introduction
A novel coronavirus was identified at the end of 2019 as the cause of a cluster of pneumonia cases in Wuhan, China. It spread rapidly throughout China and the world in a few months. Coronavirus is a member of the family Coronaviridae. "Severe acute respiratory syndrome coronavirus 2 (SARS-CoV-2)" as the name for the new virus was announced by International Committee on Taxonomy of Viruses (ICTV), in order to differentiate from the coronavirus responsible for the SARS outbreak of 2003 [2]. Currently, several coronavirus disease 2019 (COVID-19) vaccines are available, including mRNA vaccines, protein subunit vaccines, vector vaccines, etc. Temporary side effects are common, such as pain at the injection site, fever, headache or feeling tired [3]. Other serious but rare side effects include severe allergic reactions, blood clots, myocarditis, and Guillain–Barre syndrome. mRNA SARS-CoV-2 vaccines have been administered largely in the United States (US) under US Food and Drug Administration emergency use authorization [3]. The benefits of getting vaccinated are much greater than the risks of side effects caused by vaccine for most people. Regarding clinical manifestations of allergic reactions, the most common symptoms and signs are flushing, itching, urticaria, nasal discharge, nasal congestion, faintness, and syncope [4-6]. Immediate-type hypersensitivity reactions are caused by degranulation of mast cells. Most reactions are immunoglobulin E (IgE)-mediated. However, other reported allergic reactions to mRNA COVID-19 vaccines include the delayed type [1].
The majority of reported allergic reactions to mRNA COVID-19 vaccines occurred within 30 minutes of vaccination (mostly within 15 minutes) [3]. "COVID arm" is an uncommon adverse effect that can present as transient, localized erythematous rash several days after the first dose of the mRNA-1273 COVID-19 vaccine [4-6]. Baden et al. [6] reported that delayed injection site reactions occurred in 244 of the 30,420 participants (0.8%) after the first dose and in 68 participants (0.2%) after the second dose of vaccine administration. In an observational cohort study [4], delayed skin reactions (onset > two days after vaccination) were reported by 1.1% of all female who received the mRNA-1273 vaccine and 2% of females aged 31 to 45 years when stratified by age. In addition, there were no report of delayed skin reactions among the male who received the mRNA-1273 vaccine. This clinical manifestation demonstrated that women were at greater risk of adverse drug reactions than men. The difference between men and women has been proposed including decreased body mass, sex-based differences in health information seeking, sex-based differences in pharmacokinetics and pharmacodynamics. The mechanism of these delayed injection site reactions is not well known, either. The clinical suspicion of delayed-type or T-cell–mediated hypersensitivity is supported by skin biopsy specimens. Previous skin biopsy reports have shown superficial perivascular and perifollicular lymphocytic infiltrates with scattered sporadic mast cells and eosinophils [4-7]. We presented a case of “COVID arm in a woman”: a localized skin erythematous rash surrounding the injection site that arose three days after the first dose of the Moderna COVID-19 vaccine. This report also made a brief review focused on the incidence, clinical characteristics, and management of delayed skin reactions to COVID-19 mRNA vaccines.
- Discussion
Delayed injection site reactions are currently considered delayed-type (Type IV) or T-cell-mediated hypersensitivity, supported by a skin biopsy of one of the late local reactions [7], although it is unclear why the reactions tend not to recur if this is the mechanism. The mRNA vaccines contain polyethylene glycol (PEG), which may cause more inflammation or be more immunogenic. More studies are needed to determine what vaccine component drives these reactions and the immunologic mechanism [8-10]. In addition, the delayed-type skin or cutaneous reactions have been reported to be associated with other non-COVID-19 vaccinations [10].
The onset of delayed skin reactions usually begins approximately 3 days after vaccination, while one observational cohort study reported the occurrence 3-9 days (median 7) after the first dose of mRNA-1273 [4]. Jacobson, M.A. et al. reported that 13 women who received the mRNA-1273 vaccine during the first 3 weeks of SARS-CoV-2 vaccine availability presented with a pruritic rash at the injection site appearing 3-9 days after receipt of their initial dose in one hospital in the US [4]. Among these 13 females, 5 women had milder or similar reactions with earlier onset after the second dose. Only one additional woman presented with this delayed reaction after the second dose. No people reported serious adverse events or had severe symptoms that warranted further medical attention. These results suggest that delayed onset pruritic skin rashes at the injection site after mRNA-1273 SARS-CoV-2 vaccine administration lasting from days to weeks occur predominantly in women, usually do not cause serious complications or any sequelae, and should not deter receipt of the second vaccination. Therefore, neither local injection site reactions nor delayed-type hypersensitivity reactions are contraindications to subsequent vaccination. The median onset of cutaneous symptoms after the second dose was earlier than that after the first dose. Patients who have had such reactions should receive their second doses on time, without any special precautions [1]. A trend towards a lower risk of recurrent reactions was observed when the second dose was given on the opposite arm from the first dose [4]. It would be reasonable to suggest that individuals with this reaction after the first dose choose the opposite arm for their second dose. Identified sex differences in reported adverse drug reactions often referred to females. Females generally have a lower lean body mass, reduced hepatic clearance, pharmacodynamic differences in the activity of cytochrome P450 (CYP) enzymes, and metabolize drugs at different rates compared with males. Other important factors include absorption, conjugation, protein binding and renal elimination, which may all have some sex-based differences. However, how these differences lead to an increased risk of delayed skin reactions caused by the COVID-19 vaccine is not very clear [11-14]. de Vries, S. T et al. also proposed that the differences in adverse drug reactions between females and males may have various origins, including pharmacological and behavioural causes, which need to be further assessed. These differences may eventually result in sex-specific prescribing and monitoring recommendations [15]. However, the genetic effects associated with or enhance the adverse reactions after the vaccinations need to be further assessed [14,15]. We also reviewed the literature and compared clinical features, treatment, and clinical outcomes in patients with COVID-19 mRNA-1273 vaccine-induced delayed skin reactions (Table 2).
In summary, current clinical data suggest that potential recipients of the mRNA-1273 vaccine can be reassured that delayed injection site skin reactions are benign, self-limited adverse events that resolve gradually and do not lead to serious clinical problems. It is important for public health agencies to educate the public about the possibility of these delayed injection site reactions to the mRNA-1273 vaccine and reassure them. Nevertheless, more evidence is needed to characterize the epidemiology, mechanism, prevention, treatment, and why delayed injection site reactions seems to occur more frequently in females than males, even among different races.
- Conclusions
We reported a case of a delayed skin injection site reaction to the mRNA-1273 vaccine. The comprehensive mechanism is not well known, and the clinical characteristics are consistent with delayed-type hypersensitivity.
Lines 63-73. This data should be present in the “body” of the article only. To delete.
Author Reply:
Thank you for your valuable comments. We have made this correction and deleted this content immediately.
- Introduction
A novel coronavirus was identified at the end of 2019 as the cause of a cluster of pneumonia cases in Wuhan, China. It spread rapidly throughout China and the world in a few months. Coronavirus is a member of the family Coronaviridae. "Severe acute respiratory syndrome coronavirus 2 (SARS-CoV-2)" as the name for the new virus was announced by International Committee on Taxonomy of Viruses (ICTV), in order to differentiate from the coronavirus responsible for the SARS outbreak of 2003 [2]. Currently, several coronavirus disease 2019 (COVID-19) vaccines are available, including mRNA vaccines, protein subunit vaccines, vector vaccines, etc. Temporary side effects are common, such as pain at the injection site, fever, headache or feeling tired [3]. Other serious but rare side effects include severe allergic reactions, blood clots, myocarditis, and Guillain–Barre syndrome. mRNA SARS-CoV-2 vaccines have been administered largely in the United States (US) under US Food and Drug Administration emergency use authorization [3]. The benefits of getting vaccinated are much greater than the risks of side effects caused by vaccine for most people. Regarding clinical manifestations of allergic reactions, the most common symptoms and signs are flushing, itching, urticaria, nasal discharge, nasal congestion, faintness, and syncope [4-6]. Immediate-type hypersensitivity reactions are caused by degranulation of mast cells. Most reactions are immunoglobulin E (IgE)-mediated. However, other reported allergic reactions to mRNA COVID-19 vaccines include the delayed type [1].
The majority of reported allergic reactions to mRNA COVID-19 vaccines occurred within 30 minutes of vaccination (mostly within 15 minutes) [3]. "COVID arm" is an uncommon adverse effect that can present as transient, localized erythematous rash several days after the first dose of the mRNA-1273 COVID-19 vaccine [4-6]. Baden et al. [6] reported that delayed injection site reactions occurred in 244 of the 30,420 participants (0.8%) after the first dose and in 68 participants (0.2%) after the second dose of vaccine administration. In an observational cohort study [4], delayed skin reactions (onset > two days after vaccination) were reported by 1.1% of all female who received the mRNA-1273 vaccine and 2% of females aged 31 to 45 years when stratified by age. In addition, there were no report of delayed skin reactions among the male who received the mRNA-1273 vaccine. This clinical manifestation demonstrated that women were at greater risk of adverse drug reactions than men. The difference between men and women has been proposed including decreased body mass, sex-based differences in health information seeking, sex-based differences in pharmacokinetics and pharmacodynamics. The mechanism of these delayed injection site reactions is not well known, either. The clinical suspicion of delayed-type or T-cell–mediated hypersensitivity is supported by skin biopsy specimens. Previous skin biopsy reports have shown superficial perivascular and perifollicular lymphocytic infiltrates with scattered sporadic mast cells and eosinophils [4-7]. We presented a case of “COVID arm in a woman”: a localized skin erythematous rash surrounding the injection site that arose three days after the first dose of the Moderna COVID-19 vaccine. This report also made a brief review focused on the incidence, clinical characteristics, and management of delayed skin reactions to COVID-19 mRNA vaccines.
Discussion. Too long and boring. Too much abundant data.
Author Reply:
Thank you for your valuable comments. We have made this correction and shortening the content of Discussion.
Conclusions. In my opinion, basing on a single case the authors cannot made any conclusions. It should be completely deleted.
Author Reply:
Thank you for your valuable comments. We have made this correction and shortening the content of Conclusion.
- Conclusions
We reported a case of a delayed skin injection site reaction to the mRNA-1273 vaccine. The comprehensive mechanism is not well known, and the clinical characteristics are consistent with delayed-type hypersensitivity.
Minor.
Lines 51-63. Reference to every statement has to be added.
Author Reply:
Thank you for your valuable comments. We have made this correction and added the references.
- Introduction
A novel coronavirus was identified at the end of 2019 as the cause of a cluster of pneumonia cases in Wuhan, China. It spread rapidly throughout China and the world in a few months. Coronavirus is a member of the family Coronaviridae. "Severe acute respiratory syndrome coronavirus 2 (SARS-CoV-2)" as the name for the new virus was announced by International Committee on Taxonomy of Viruses (ICTV), in order to differentiate from the coronavirus responsible for the SARS outbreak of 2003 [2]. Currently, several coronavirus disease 2019 (COVID-19) vaccines are available, including mRNA vaccines, protein subunit vaccines, vector vaccines, etc. Temporary side effects are common, such as pain at the injection site, fever, headache or feeling tired [3]. Other serious but rare side effects include severe allergic reactions, blood clots, myocarditis, and Guillain–Barre syndrome. mRNA SARS-CoV-2 vaccines have been administered largely in the United States (US) under US Food and Drug Administration emergency use authorization [3]. The benefits of getting vaccinated are much greater than the risks of side effects caused by vaccine for most people. Regarding clinical manifestations of allergic reactions, the most common symptoms and signs are flushing, itching, urticaria, nasal discharge, nasal congestion, faintness, and syncope [4-6]. Immediate-type hypersensitivity reactions are caused by degranulation of mast cells. Most reactions are immunoglobulin E (IgE)-mediated. However, other reported allergic reactions to mRNA COVID-19 vaccines include the delayed type [1].
References
- Blumenthal, K.G.; Freeman, E.E.; Saff, R.R.; Robinson, L.B.; Wolfson, A.R.; Foreman, R,K. Hashimoto, D.; Banerji, A.; Li, L.; Anvari, S.; Shenoy, E.S. Delayed Large Local Reactions to mRNA-1273 Vaccine against SARS-CoV-2. N Engl J Med 2021, 384, 1273-1277.
- Khan, T., Jamal, S.M. SARS-CoV-2 nomenclature: viruses, variants and vaccines need a standardized naming system. Future Virol 2021 Oct, 10.2217/fvl-2021-0198. doi: 10.2217/fvl-2021-0198.
- CDC COVID-19 Response Team; Food and Drug Administration.Allergic Reactions Including Anaphylaxis After Receipt of the First Dose of Moderna COVID-19 Vaccine – United States, December 21, 2020-January 10, 2021. MMWR Morb Mortal Wkly Rep 2021, 70, 125-129. doi: 10.15585/mmwr.mm7004e1.
- Jacobson, M.A.; Zakaria, A.; Maung, Z.; Hart, C.; McCalmont, T.H.; Fassett, M.; Amerson, E. Incidence and Characteristics of Delayed Injection Site Reaction to the mRNA-1273 SARS-CoV2 Vaccine (Moderna) in a Cohort of Hospital Employees. Clin Infect Dis 2022, 74, 591-596. doi: 10.1093/cid/ciab518.
- Wei, N.; Fishman, M.; Wattenberg, D.; Gordon. M.; Lebwohl, M. "COVID arm": A reaction to the Moderna vaccine. JAAD Case Rep 2021, 10, 92-95. doi: 10.1016/j.jdcr.2021.02.014.
- Baden, L.R.; El Sahly, H.M.; Essink, B.; Kotloff, K.; Frey, S.; Novak, R.; Diemert, D,; Spector, S.A.; Rouphael, N.; Creech, CB.; McGettigan, J.; Khetan, S.; Segall, N.; Solis, J.; Brosz, A.; Fierr, C.; Schwartz, H.; Neuzil, K.; Corey, L.; Gilbert, P.; Janes, H.; Follmann, D.; Marovich, M.; Mascola, J.; Polakowski, L.; Ledgerwood, J.; Graham, B.S.; Bennett, H.; Pajon. R.; Knightly, C.; Leav, B.; Deng, W.; Zhou, H.; Han, S.; Ivarsson, M.; Miller, J.; Zaks, T.; COVE Study Group. Efficacy and Safety of the mRNA-1273 SARS-CoV-2 Vaccine. N Engl J Med 2021, 384, 403-416. doi: 10.1056/NEJMoa2035389.
Last, we are deeply honored by the time and effort you spent reviewing this manuscript. In reviewing and revising our manuscript, we are motivated to read more and thus learn more from your criticisms.
